# A new type of radial basis functions for problems governed by partial differential equations

Jie Liu[1], Fuzhang Wang[2]*, Sohail Nadeem[3]

1 School of Mathematical Sciences, Huaibei Normal University, Huaibei, China, 2 College of Education, Nanchang Normal College of Applied Technology, Nanchang, China, 3 Department of Mathematics, Quaid-i-Azam University, Islamabad, Pakistan

* wangfuzhang1984@163.com

**Data Availability Statement:** All relevant data are within the paper.

**Funding:** "The work was supported by the Natural Science Research Project for Huaibei Normal University (2023ZK034), the Research Project of

## Abstract

The aim of this paper is to introduce a novel category of radial basis functions that incorporate smoothing techniques. Initially, we employ the power augmented and shape parameter schemes to create the radial basis functions. Subsequently, we apply the newly-constructed radial basis functions using the traditional collocation method and singular values decomposition algorithm to solve the corresponding linear system equations. Finally, we analyze several pairs of radial basis functions in depth to address physical problems linked to thermal science that are governed by partial differential equations. The numerical results demonstrate that the radial basis functions constructed using the power augmented and shape parameter schemes exhibit remarkable performance.

## 1. Introduction

The radial basis function (RBF) is a powerful tool in numerical simulations of engineering problems related to thermal science. Due to the significant advantage of dependence of the Euclidean distance variable, RBFs have been popularly employed to approximate the field variable. The Euclidean distances are easy to be computed in arbitrary space dimensions. Therefore, RBFs can be used to solve higher-dimensional problems with easy implementation. Another appealing feature of RBFs is that the usage of them is fully meshfree or meshless, i.e., only collocation points are used.

Currently, there are multiple types of RBF-based meshless methods that are being used for solving elliptical problems. Among these methods, the most commonly used one is the pure collocation method, which is also referred to as the direct collocation method. A software suite consisting of 17 MATLAB functions for solving differential equations by the spectral collocation method is presented. It includes functions for computing derivatives of arbitrary order corresponding to Chebyshev, Hermite, Laguerre, Fourier, and sinc interpolants [1]. The essentials of spectral collocation methods with the aid of forty short Matlab R programs, or "M-files" can be referred to [2]. In the collocation approach, the numerical solution is approximated directly using a combination of RBFs with unknown coefficients. The basic idea behind this

Science and Technology Plan of Jiangxi Provincial Department of Education (No. GJJ2203205), the Natural Science Foundation of Jiangxi Province (Project No. 20224BAB201018)".

**Competing interests:** The authors have declared that no competing interests exist.

method is to select a set of collocation points where the exact solution of the differential equation is already known, and then to use these points to determine the unknown coefficients of the RBF approximation. By doing so, the RBF-based meshless method can effectively solve elliptical problems without the need for a structured mesh, which can be time-consuming and computationally expensive to generate. The famous Kansa's method [3], the method of fundamental solution [4], the boundary knot method [5] and the modified method of fundamental solutions [6] are typical examples. It is worth pointing out that the above-mentioned methods are based on the global interpolation of RBFs. The expansion coefficients instead of physical variables are used as the basic unknowns, while the system matrix is generally full and unsymmetrical. Some improved technologies are developed to overcome these drawbacks. For Hermite-Birkhoff interpolation of scattered multidimensional data by radial basis function, existence and characterization theorems and a variational principle are proved in [7]. A new class of positive definite and compactly supported radial functions, which consist of a univariate polynomial within their support, is constructed in [8]. The advantages of simplicity and effectiveness make those meshless methods based on globally supported RBFs attractive to engineers. For brevity, we employ the traditional collocation method to implement our computation in this paper.

The proper usage of RBF-based meshless methods depends on the construction of RBFs. The most of popular RBFs are created based on the second Green identity [9], it has made some significant advances in the boundary-only and domain-type RBF techniques. For instance, the method of Multiquadrics, which utilizes the fundamental solutions of the 1D Laplacian operator, while the method of thin plate splines is based on the fundamental solutions of the 2D Laplacian operator. Because the fundamental solutions have singularity at origin, how to construct the smoothing kernel RBFs becomes a considerable issue. As is known to all, there are two common schemes to obtain the smoothing kernel RBFs. The first is to apply $r^{2k}$ augmented term to enhance the smoothness which ensures sufficient degree of differential continuity, where $k$ is a positive integer. The TPS is a notable example. The second is to replace the distance variable $r$ used in the fundamental solution by $\sqrt{r^2 + c^2}$, where c is a positive parameter. The MQs is a typical one. In this paper, three pairs of RBFs constructed by means of these two smoothing schemes are compared in solving the complicated elliptical problems from the aspects of accuracy, convergence and stability to parameters. On the one hand, RBFs created by using the PAS are piecewise smooth and the degree of smoothness is controlled by the integer $k$, while RBFs created by using the SPS are infinitely smooth. On the other hand, unlike MQs, the SEF has an inherent feature, that is, it can produce a diagonal dominant system matrix. The diagonal element is the largest in each row.

The structure of the paper is arranged as follows. In Section 2, the tested problems are briefly introduced. Section 3 describes the traditional collocation method and gives three pairs of RBFs constructed by means of two smoothing approaches. Followed by Section 4, numerical comparisons related to accuracy, convergence and stability to parameters are carried out, and finally Section 5 concludes some comments on these two smoothing schemes and some future directions.

## 2. Problem description

It is widely recognized that a significant number of physical phenomena in thermal science can be represented and studied through the use of partial differential equations (PDEs) that are defined in an open bounded domain $\Omega \subset R^d$ [10–12]

$$Lu(X) = f(X) \tag{1}$$

where $d$ denotes the dimensionality of the space, $L(\bullet)$ represents the general linear operator and $u$ is the desired field, $f(X)$ denotes the internal forcing function. The spatial coordinate $X = (x_1, x_2, \ldots, x_d) \in \Omega \subseteq R^d$ is used to determine the position of a point.

For the typical Poisson problems, Helmholtz problems and modified Helmholtz problems, the linear operator $L(\bullet)$ is expressed as

$$L(u) = \begin{cases} \displaystyle\sum_{i=1}^{d}\sum_{j=1}^{d} k_{ij}\frac{\partial^2 u}{\partial x_i \partial x_j} & \text{for Poisson case} \\[2ex] \displaystyle\sum_{i=1}^{d}\sum_{j=1}^{d} k_{ij}\frac{\partial^2 u}{\partial x_i \partial x_j} + \lambda^2 u & \text{for Helmholtze case} \\[2ex] \displaystyle\sum_{i=1}^{d}\sum_{j=1}^{d} k_{ij}\frac{\partial^2 u}{\partial x_i \partial x_j} - \lambda^2 u & \text{for modified Helmholtze case} \end{cases} \qquad (2)$$

where $\mathbf{K} = [k_{ij}]$ represents a material tensor that is assumed to have a positive-definite value, while $\lambda$ is used to denote the wave number.

To make Eq (1) solvable, the boundary conditions on a piecewise smooth boundary $\Gamma$ should be considered

$$\begin{cases} u(X) = \bar{u}(X) & X \in \Gamma_1 \\ q(X) = \bar{q}(X) & X \in \Gamma_2 \end{cases} \qquad (3)$$

where $\bar{u}$ and $\bar{q}$ are provided on the Dirichlet boundary $\Gamma_1$ and Neumann boundary $\Gamma_2$, respectively. The boundary flux arising in the boundary conditions is defined as $q = \displaystyle\sum_{i=1}^{d}\sum_{j=1}^{d} k_{ij}\frac{\partial u}{\partial x_j}n_i$ with $n_i$ denoting components of the unit outward normal vector $\mathbf{n}$ to the boundary $\Gamma$.

## 3. Collocation method

### 3.1 Traditional collocation method

The traditional collocation method is a direct RBF-based method for elliptic problems. Numerical experiments show that this type of methods work well for most cases [13, 14]. The collocation method offers several benefits, particularly in terms of its straightforwardness and practicality in application. Here, we employ the collocation method to carry out our investigations.

Suppose we have an arbitrary point $X$ within the domain, and we approximate the solution $u$ using RBFs

$$u(X) = \sum_{j=1}^{N+M} c_j \phi_j(X) \qquad (4)$$

where $N$ and $M$ are the number of collocation points along the boundary and over the interior, respectively. $\phi_j(X) = \phi(r_j)$ represents a globally supported RBF and $r_j = \sqrt{(X - X_j)^T (X - X_j)}$ the Euclidian distance defined between the field point $X$ and the central point $X_j$.

Collocating the boundary data at the boundary points and the governing equation at the interior points lead to the following linear system of equations

$$\sum_{k=1}^{N+M} c_k [L(\phi_k(X_s))] = f(X_s), \; s = 1, \ldots, M \tag{5}$$

$$\sum_{k=1}^{N+M} c_k \phi_k(X_l) = \bar{u}(X_l), \;\; l = 1, \ldots, N_1 \tag{6}$$

$$\sum_{k=1}^{N+M} c_k \left[ \sum_{i=1}^{d} \sum_{j=1}^{d} k_{ij} \frac{\partial \phi_k(X_l)}{\partial x_j} n_i \right] = \bar{q}(X_l), \; l = 1, \ldots, N_2 \tag{7}$$

$$\text{with } L(\phi_k(X_s)) = \begin{cases} \displaystyle\sum_{i=1}^{d}\sum_{j=1}^{d} k_{ij} \frac{\partial^2 \phi_k(X_s)}{\partial x_i \partial x_j} & \text{for Poisson case} \\[2em] \displaystyle\sum_{i=1}^{d}\sum_{j=1}^{d} k_{ij} \frac{\partial^2 \phi_k(X_s)}{\partial x_i \partial x_j} + \lambda^2 \phi_k(X_s) & \text{for Helmholtze case} \\[2em] \displaystyle\sum_{i=1}^{d}\sum_{j=1}^{d} k_{ij} \frac{\partial^2 \phi_k(X_s)}{\partial x_i \partial x_j} - \lambda^2 \phi_k(X_s) & \text{for modified Helmholtze case} \end{cases},$$

where $N_1$ and $N_2$ ($N = N_1 + N_2$) are number of collocation points on boundary $\Gamma_1$ and $\Gamma_2$, respectively.

## 3.2 Tested radial basis functions

Here, three pairs of RBFs are listed in Table 1 to show the different smoothing schemes. For convenience, we name the scheme multiplying $r^{2k}$ as the power augmented scheme (PAS), while another scheme introducing shape parameter $c$ is named as the shape parameter scheme (SPS).

For the above RBFs, there are two aspects which should be mentioned. On the one hand, RBFs created by using the PAS are piecewise smooth and the degree of smoothness is controlled by the integer $k$, while RBFs created by using the SPS are infinitely smooth. On the other hand, unlike MQs, the SEF has an inherent feature, that is, it can produce a diagonal dominant system matrix. The diagonal element is the largest in each row.

Take the MQs for example, we can get the derivatives of MQs in the following expression.

$$\phi(r) = \sqrt{r^2 + c^2}, \; r = \|X - Y\| = \sqrt{(x_1 - x_2)^2 + (y_1 - y_2)^2}, \; r_x = \frac{x_1 - x_2}{r},$$

$$\phi_x(r) = \frac{1}{2}(r^2 + c^2)^{-1/2}.2r.r_x = (r^2 + c^2)^{-1/2}.(x_1 - x_2),$$

**Table 1. Smoothing RBFs created by PAS and SPS.**

| PAS | | SPS | |
|---|---|---|---|
| Power spline (PS) | $r^{2k+1}$ | Multiquadrics (MQs) | $\sqrt{r^2 + c^2}$ |
| Thin plate splines (TPS) | $r^{2k}\ln(r)$ | Shifted logarithm function (SLF) | $\ln\sqrt{r^2 + c^2}$ |
| Powered exponential function (PEF) | $r^{2k}e^{-r}$ | Shifted exponential function (SEF) | $e^{-\sqrt{r^2+c^2}}$ |

$$\varphi_y(r) = \frac{1}{2}(r^2 + c^2)^{-1/2}.2r.r_y = (r^2 + c^2)^{-1/2}.(y_1 - y_2),$$

$$\varphi_{xx}(r) = (r^2 + c^2)^{-1/2}\left[1 - \frac{(x_1 - x_2)^2}{r^2 + c^2}\right],$$

$$\varphi_{yy}(r) = (r^2 + c^2)^{-1/2}\left[1 - \frac{(y_1 - y_2)^2}{r^2 + c^2}\right].$$

In the next section, these RBFs will be tested to investigate their behaviors involving accuracy, convergence and sensitivity to parameter in solving problems governed by partial differential equations.

## 4. Numerical experiments

Before carrying out our numerical experiments, we provide a uniform computing platform. Because of the typically large condition number found in RBF-based techniques, the approach of decomposing singular values is utilized to solve linear system equations. When the condition number of a coefficient matrix is infinite, it is considered singular, and when the condition number is excessively large, it is considered ill-conditioned. To ensure the proper application of SVD, it is recommended to disregard the small singular values, according to [15]. Here, we set $10^{-12}$ as the threshold for singular values allowed to be positive. For the sake of convenience, all computations are investigated on a unit square domain with only Dirichlet boundary condition involved. Collocation points are uniformly distributed on the physical domain.

The average relative error Arerr($u$) is given as follows

$$\text{Arerr}(u) = \sqrt{\frac{\sum_{j=1}^{L}(u_j - \bar{u}_j)^2}{\sum_{j=1}^{L}(u_j)^2}} \tag{8}$$

where $u_j$ and $\bar{u}_j$ are the analytical and numerical results at test points of interest, respectively. $L$ denotes the total number of tested points of interest. Here, $L = 10000$ is chosen in the following tested cases. In addition, the parameter $k = 3$ is settled.

### 4.1 2D Poisson problems

First, we consider the case of 2D Poisson equation

$$\nabla^2 u = \frac{-751\pi^2}{144}\sin\frac{\pi x}{6}\sin\frac{7\pi x}{4}\sin\frac{3\pi y}{4}\sin\frac{5\pi y}{4}$$
$$+ \frac{7\pi^2}{12}\cos\frac{\pi x}{6}\cos\frac{7\pi x}{4}\sin\frac{3\pi y}{4}\sin\frac{5\pi y}{4} + \frac{15\pi^2}{8}\sin\frac{\pi x}{6}\sin\frac{7\pi x}{4}\cos\frac{3\pi y}{4}\cos\frac{5\pi y}{4} \tag{9}$$

The particular solution satisfying Eq (9) is selected to be

$$u(x, y) = \sin\frac{\pi x}{6}\sin\frac{7\pi x}{4}\sin\frac{3\pi y}{4}\sin\frac{5\pi y}{4}. \tag{10}$$

For the same collocation point parameter N = 52, Fig 1 depicts the variations of Arerr and the condition numbers for the MQs, SLF and SEF as the shape parameter c varies in the range of (0,3]. We can see that these RBFs have some common characters, the variation curves for the MQs, SLF and SEF are similar for different interior collocation numbers M = 25, 64, 100, 144. This reveals that the proposed SLF and SEF have the same merits with MQs. At the same time, the oscillation phenomenon of the condition numbers for SLF is a little better than those of the MQs and SEF. On the one hand, the choice of the shape parameter c obviously influences the solution accuracy. For instance, the parameter $c$ in [0.1,0.9], [0.4,1.2] and [0.1,1.0] can reach relatively stable convergence for MQs, SLF and SEF, respectively, and the highest accuracy is usually obtained around $c = 0.9$. On the other hand, the Arerr decreases as the parameter c increases, while the corresponding condition number behaves contrarily to the increase of $c$. When the parameter $c \geq 1$, the condition number begins to oscillate around a larger value $10^{20}$, that is to say, the system matrix becomes ill-conditioning, which may be the reason for the accuracy reduction. We can conclude that the shape parameter can be chosen around 1 for 2D passion equation problems.

## 4.2 2D anisotropic Helmholtz problems

We consider the following 2D anisotropic Helmholtz equation

$$\frac{\partial^2 u}{\partial x_1{}^2} + \frac{\partial^2 u}{\partial x_1 \partial x_2} + \frac{\partial^2 u}{\partial x_2{}^2} + \lambda^2 u = 0. \tag{11}$$

The particular solution is

$$u = \sin\left(5\sqrt{3}x\right) + \cos\left[10\left(y - \frac{x}{2}\right)\right] \tag{12}$$

for the case of $\lambda = 5\sqrt{3}$.

Numerical results (Fig 2) show the curves of Arerr and the condition numbers for the MQs, SLF and SEF as the shape parameter c varies in the range of (0,3], similar conclusions obtained in example 4.1 can also be drawn. The significant difference is that the curve of Arerr become more mild, which means that the convergent interval maybe enlarge. The convergent ranges of the parameter $c$ are [0.2,1.4], [0.4,1.5] and [0.2,1.5] for MQs, SLF and SEF, respectively, and the highest accuracy also can be obtained around $c = 0.9$. However, due to the strong oscillation of solution in the domain, the numerical accuracy is one order of magnitude smaller than results shown in example 4.1.

## 4.3 2D anisotropic modified Helmholtz problem

The modified Helmholtz equation is a mathematical tool frequently utilized to simulate and understand the process of heat conduction, as well as chemical reactions in engineering. This equation is commonly referred to as the diffusion-reaction equation due to its ability to model the transfer of substances between different media. Considered a 2D anisotropic modified Helmholtz equation

$$5\frac{\partial^2 u}{\partial x^2} + 4\frac{\partial^2 u}{\partial x \partial y} + \frac{\partial^2 u}{\partial y^2} - \lambda^2 u = 0. \tag{13}$$

It is apparent that the following solution

$$u = e^{(3x-10y)} \tag{14}$$

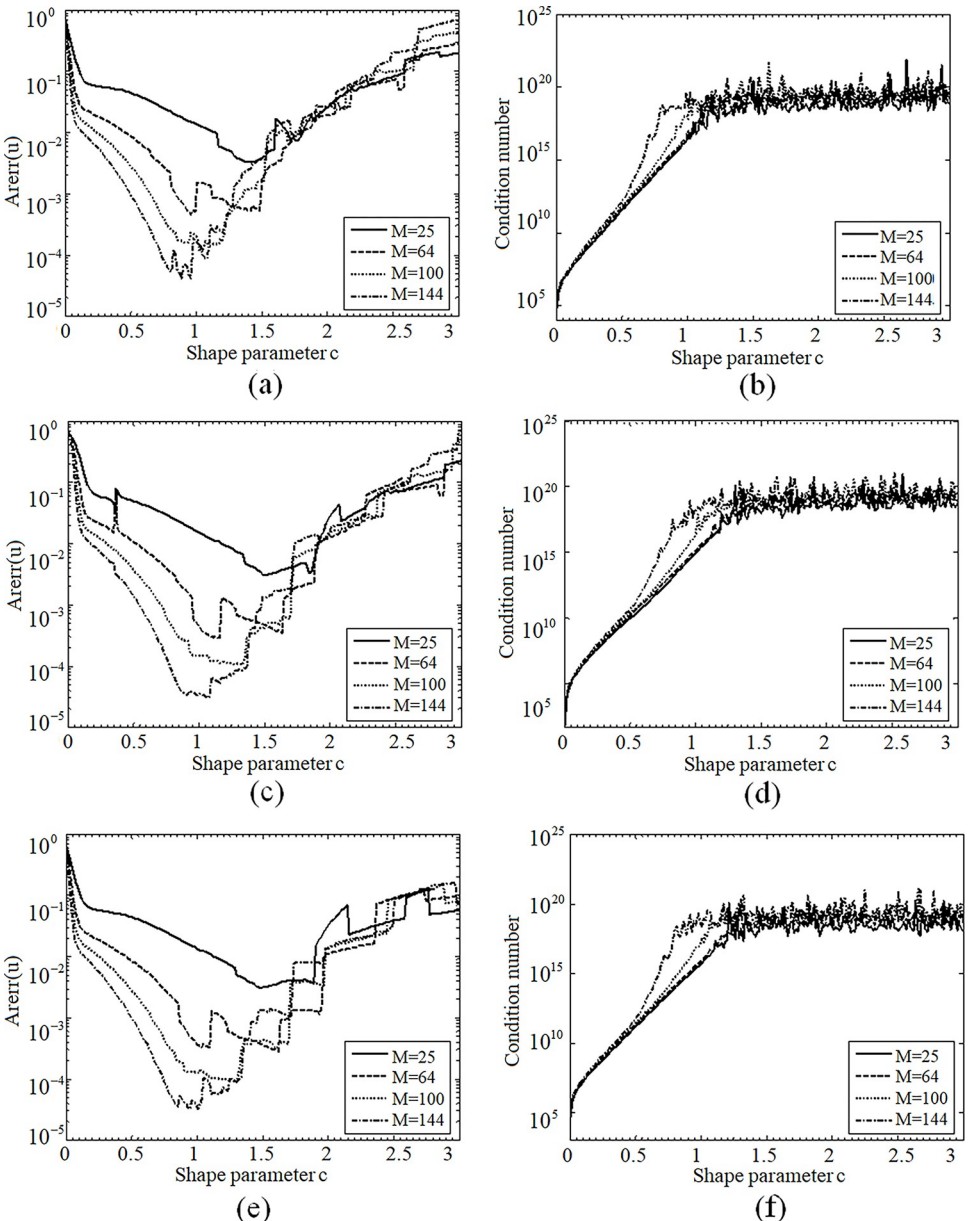

**Fig 1.** (a) Curves of Arerr number for MO, (b) Curves of condition number for MO, (c)Curves of Arerr number for SLF, (d) Curves of condition number for SLF, (e) Curves ofArerr numberfor SEF, (f) Curves of condition numberfor SEF.

exactly satisfies the 2D modified Helmholtz Eq (13).

Fig 3 give the curves of Arerr and condition number for the MQs, SLF and SEF as the shape parameter c varies in the range of (0,3]. The better accuracy and good convergence can be reached at the interval of [0.75,0.85], [0.9,1.1] and [0.8,1.1] for the MQs, SLF and SEF, respectively. The fact that the MQs has the smallest range means the sharp variation of accuracy and is disadvantageous to the practical computation.

For the same collocation point parameter N = 52, results in Fig 4 show that all three RBFs PS, TPS and PEF have similar convergent rate, while the PS has the highest accuracy among all piecewise smoothed RBFs.

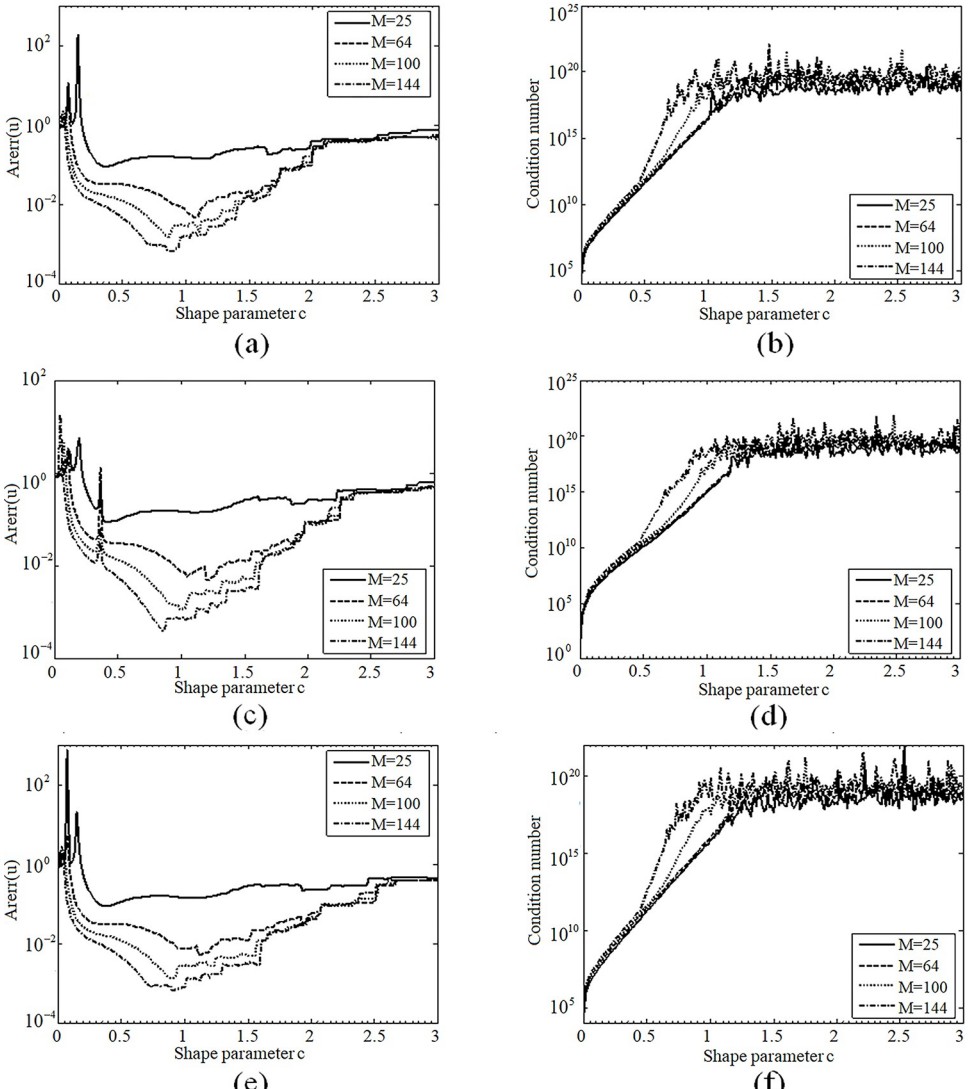

**Fig 2.** (a) Curves of Arerr for MO, (b) Curves of condition number for MO, (c) Curves ofArerr for SLF, (d) Curves of condition number for SLF, (e) Curves of Arerr for SEF, (f) Curves of condition number for SEF.

### 4.4 Discussions

Compared to RBFs created by means of the SPS, piecewise smoothed PS, TPS and PEF can keep good stability and convergence. The related results for the previous three examples can be found in Fig 4. Fig 4(A) and 4(B) is for example 4.1, Fig 4(C) and 4(D) is for example 4.2 and Fig 4(E), 4(f) is for example 4.3. At the same time, we can see that the PS has the highest accuracy and the rapidest convergence among all piecewise smoothed RBFs, while the condition number shows correspondingly contrary phenomenon.

## 5. Conclusions

By comparing the two different smoothing strategies in constructing RBFs, we find that the PAS and the SPS have distinctive features in the aspects of numerical accuracy, stability to parameters and convergence. In summary, we can draw the following conclusions:

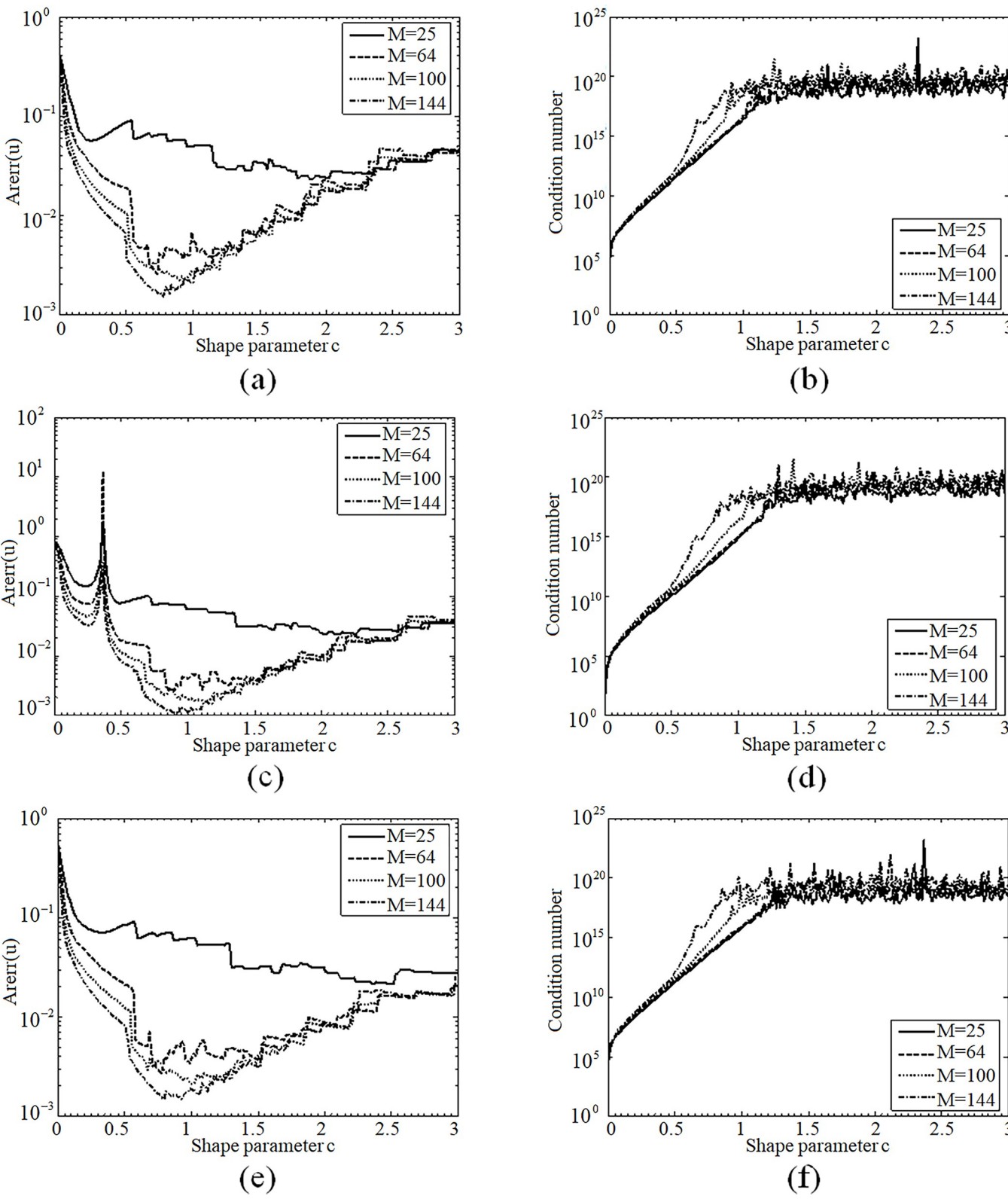

**Fig 3.** (a) Distribution of Arerr for MO, (b) Distribution of condition number for MO, (c) Distribution of Arerr for SLF, (d) Distribution of condition number for SLF, (e) Distribution of Arerr for SEF, (f) Distribution of condition number for SEF.

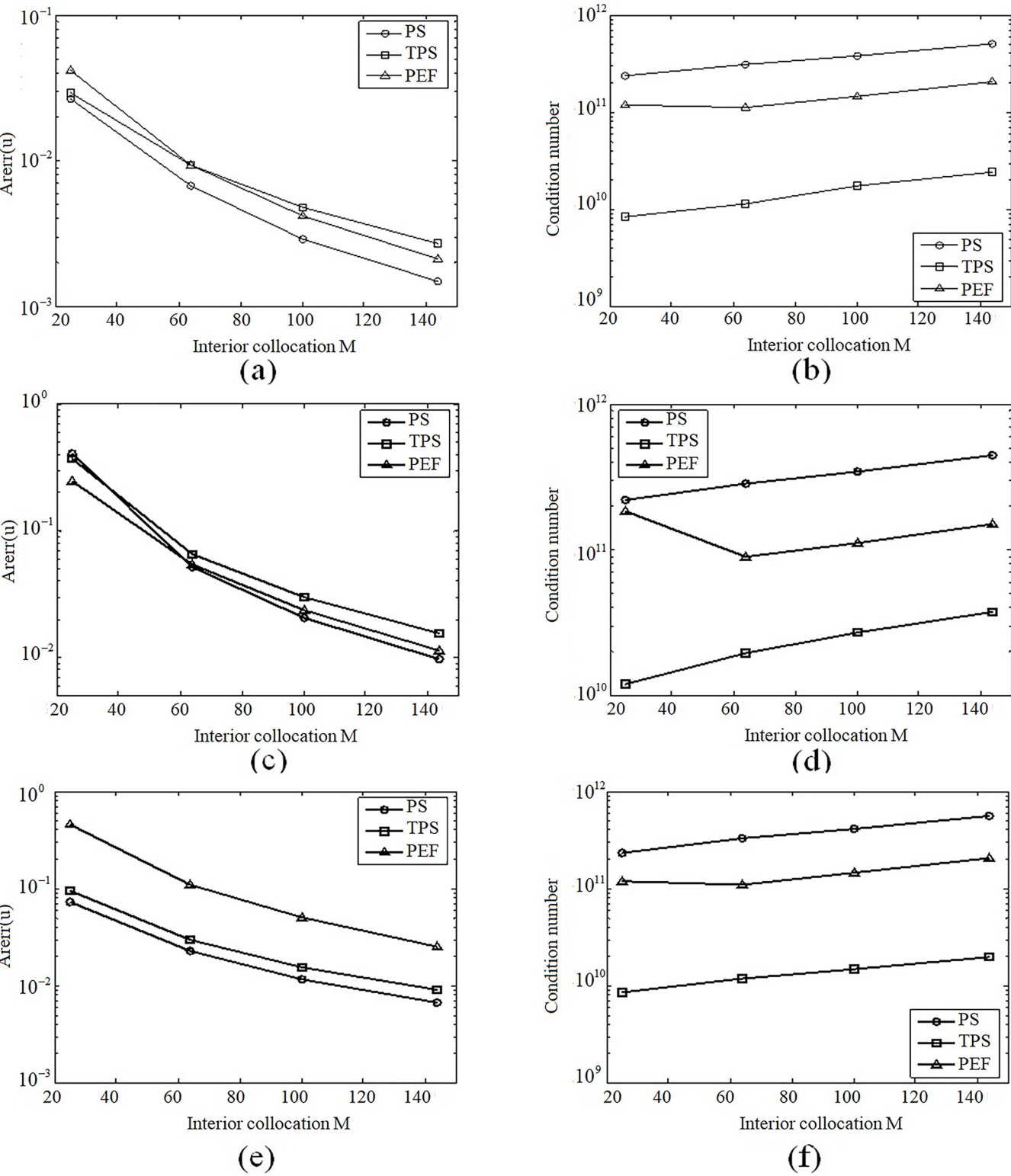

**Fig 4.** (a) Curves of Arerr for PS, (b) Curves of condition number for PS, (C) Curves of Arerrfor TPS, (d) Curves of condition number for TPS, (e) Curves of Arerr for PEF, (f) Curves ofconditionnumberfor PEF.

1. These newly-constructed RBFs have some common characters, the variation curves for the MQs, SLF and SEF are similar for different interior collocation numbers. This reveals that the proposed SLF and SEF have the same merits with MQs.

2. The oscillation phenomenon of the condition numbers for SLF is a little better than those of the MQs and SEF.

3. In general, RBFs created by means of the PAS have lower accuracy than RBFs created by means of the SPS, while the former have better stability on convergence.

4. The PS has the highest accuracy accompanied with the larger condition number among all piecewise smoothed RBFs.

5. The shape parameter $c$ severely influences the accuracy and convergence, so it should be carefully selected.

6. MQs is not always the best choice for all computations tested above, SLF and SEF can replace it in some cases.

The proposed RBFs are promising in developing algorithms for large-scale problems [16–18] and problems governed by fractional equations [19].

## Author Contributions

**Conceptualization:** Jie Liu.

**Formal analysis:** Fuzhang Wang, Sohail Nadeem.

**Investigation:** Fuzhang Wang, Sohail Nadeem.

**Methodology:** Fuzhang Wang.

**Writing – original draft:** Fuzhang Wang, Sohail Nadeem.

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
