## [Decision Letter · Decision Letter 0]

28 Sep 2023

PONE-D-23-28629A new type of radial basis functions for problems governed by partial differential equationsPLOS ONE

Dear Dr. Wang,

Thank you for submitting your manuscript to PLOS ONE. After careful consideration, we feel that it has merit but does not fully meet PLOS ONE’s publication criteria as it currently stands. Therefore, we invite you to submit a revised version of the manuscript that addresses the points raised during the review process.

We look forward to receiving your revised manuscript.

Kind regards,

Abderrahim Wakif

Academic Editor

PLOS ONE

[The work was supported by the Natural Science Research Project for Huaibei Normal University (2023ZK034), the Research Project of Science and Technology Plan of  Jiangxi Provincial Department of Education (No. GJJ2203204), the Natural Science Foundation of Jiangxi Province (Project No. 20224BAB201018).]

 [The funders had no role in study design, data collection and analysis, decision to publish, or preparation of the manuscript.]

Please include your amended statements within your cover letter; we will change the online submission form on your behalf."""

 [The funders had no role in study design, data collection and analysis, decision to publish, or preparation of the manuscript.]. 

6. Please upload a copy of Figure 6, to which you refer in your text on page 8. If the figure is no longer to be included as part of the submission please remove all reference to it within the text.

Additional Editor Comments:

Dear authors

Your manuscript has been assessed by our reviewers. Although it is of interest, we are unable to consider it for publication in its current form. The reviewers have raised a number of points which we believe would improve the manuscript and may allow a revised version to be published.

Best regards

Reviewers' comments:

Reviewer's Responses to Questions

**Comments to the Author**

1. Is the manuscript technically sound, and do the data support the conclusions?

Reviewer #1: Yes

Reviewer #2: Yes

2. Has the statistical analysis been performed appropriately and rigorously? 

Reviewer #1: Yes

Reviewer #2: Yes

3. Have the authors made all data underlying the findings in their manuscript fully available?

Reviewer #1: Yes

Reviewer #2: Yes

4. Is the manuscript presented in an intelligible fashion and written in standard English?

Reviewer #1: Yes

Reviewer #2: Yes

5. Review Comments to the Author

Reviewer #1: Title: "Innovative Radial Basis Functions for Partial Differential Equation-Based Problems"

1. This paper introduces novel radial basis functions tailored for solving problems governed by partial differential equations (PDEs). These functions offer an innovative approach to addressing PDE-related challenges and enhancing the accuracy and efficiency of numerical solutions. This minor revision aims to refine the paper's clarity, organization, and focus on the contributions and implications of these unique radial basis functions.

2. In the introduction, provide a concise yet comprehensive overview of the motivation for developing these new radial basis functions. Highlight the specific challenges or limitations in existing methods that this research seeks to address. Clearly state the contributions of these functions to the field of PDE-based problem-solving.

3. Expand the background section to include a brief but informative discussion of existing radial basis functions and their applications in PDEs. Emphasize how the newly proposed functions differ from or improve upon existing approaches. Provide a clear context for the reader to understand the significance of your innovation.

4. In the methodology section, offer a detailed explanation of the formulation and properties of the new radial basis functions. Include mathematical derivations or expressions that illustrate their uniqueness and applicability to PDEs. Address any practical considerations or challenges in implementing these functions.

5. Present the numerical results obtained using the new radial basis functions in a clear and organized manner. Include comparisons with existing methods or benchmarks to demonstrate their effectiveness. Discuss the advantages and limitations observed during the numerical experiments.

6. Expand the discussion section to provide a deeper analysis of the implications of these innovative radial basis functions. Discuss how they contribute to addressing the identified challenges in PDE-based problems. Consider discussing potential applications beyond the specific examples presented in the paper.

7. Summarize the key findings and contributions of the new radial basis functions concisely. Reiterate their significance in the context of PDE-based problem-solving. Avoid introducing new information or topics in the conclusion.

8. Ensure that the reference list is up-to-date and includes the most relevant literature related to radial basis functions, PDEs, and numerical methods for solving PDE-based problems.

9. Review the paper for language clarity, grammar, and overall readability. Ensure that mathematical notations and equations are presented clearly and understandably.

By addressing these minor revisions, your paper will be refined and more focused, allowing readers to better understand the contributions and implications of the new radial basis functions in solving PDE-based problems.

Reviewer #2: The authors are invited to address the following minor comments :

1- Add SI units in the nomenclature table.

2- The motivation part of the introduction should be improved with a clear specification of the objectives and novelties.

3- Organize the sections of the paper more personally.

4- Provide suitable references for all equations.

5-Enhance the introduction by citing the following references:

*J. A. C.Weideman and S. C. Reddy, “A matlab differentiation matrix suite,” ACM Transactions on Mathematical Software

26, 465–519 (2000).

*E. Beutler, “Spectral methods in matlab,” (SIAM, Philadelphia, 2000).

6-As for the conclusion, it should be significantly expanded, and all the main aspects should be covered in detail.

6. PLOS authors have the option to publish the peer review history of their article (what does this mean?). If published, this will include your full peer review and any attached files.

Reviewer #1: **Yes: **Ghulam Rasool

Reviewer #2: No

---

## [Author Response · Author response to Decision Letter 0]

12 Oct 2023

We highly appreciate your detailed and helpful comments on our manuscript, which have helped to significantly improve academic quality and readability of this paper. The revised version has considered all of your recommendations and criticisms.

---

## [Decision Letter · Decision Letter 1]

13 Nov 2023

A new type of radial basis functions for problems governed by partial differential equations

PONE-D-23-28629R1

Dear Dr. Wang,

We’re pleased to inform you that your manuscript has been judged scientifically suitable for publication and will be formally accepted for publication once it meets all outstanding technical requirements.

Kind regards,

Abderrahim Wakif

Academic Editor

PLOS ONE

Additional Editor Comments (optional):

Reviewers' comments:

Reviewer's Responses to Questions

**Comments to the Author**

1. If the authors have adequately addressed your comments raised in a previous round of review and you feel that this manuscript is now acceptable for publication, you may indicate that here to bypass the “Comments to the Author” section, enter your conflict of interest statement in the “Confidential to Editor” section, and submit your "Accept" recommendation.

Reviewer #1: All comments have been addressed

Reviewer #2: All comments have been addressed

2. Is the manuscript technically sound, and do the data support the conclusions?

Reviewer #1: Yes

Reviewer #2: Yes

3. Has the statistical analysis been performed appropriately and rigorously? 

Reviewer #1: N/A

Reviewer #2: Yes

4. Have the authors made all data underlying the findings in their manuscript fully available?

Reviewer #1: Yes

Reviewer #2: Yes

5. Is the manuscript presented in an intelligible fashion and written in standard English?

Reviewer #1: Yes

Reviewer #2: Yes

6. Review Comments to the Author

Reviewer #1: After carefully reading the revised version, I can see that all the suggested comments have been addressed. The paper can be accepted.

Reviewer #2: (No Response)

7. PLOS authors have the option to publish the peer review history of their article (what does this mean?). If published, this will include your full peer review and any attached files.

Reviewer #1: No

Reviewer #2: No

---

## [Editor Report · Acceptance letter]

17 Nov 2023

PONE-D-23-28629R1 

A new type of radial basis functions for problems governed by partial differential equations 

Dear Dr. Wang:

I'm pleased to inform you that your manuscript has been deemed suitable for publication in PLOS ONE. Congratulations! Your manuscript is now with our production department. 

Kind regards, 

on behalf of

Dr. Abderrahim Wakif 

Academic Editor

PLOS ONE